



# 1  Piloting novel multi-centennial palaeoclimate records from mainland

# 2  southeast Australia

Jacinda A. O'Connor[1], Benjamin J. Henley[1,2,3,4], Matthew T. Brookhouse[5], Kathryn J. Allen[6,7,8]

[1]School of Earth, Atmosphere and Environment, Monash University, Clayton, 3800, Australia

[2]Department of Infrastructure Engineering, University of Melbourne, Parkville, 3010, Australia

[3]Securing Antarctica's Environmental Future (SAEF), Monash University, Clayton, 3800, Australia

[4]Australian Research Council Centre of Excellence for Climate Extremes (CLEX), Monash University, Clayton, 3800,
Australia

[5]Fenner School of Environment and Society, Australian National University, Acton 2600, Australia

[6]School of Geography, Planning, and Spatial Sciences, University of Tasmania, Sandy Bay 7005, Australia

[7] School of Ecosystem and Forest Sciences, University of Melbourne, Richmond 3121, Australia

[8] Centre of Excellence for Australian Biodiversity and Heritage, University of New South Wales, Kensington, 2052, Australia

*Correspondence to*: Jacinda A. O'Connor (jacinda.oconnor@gmail.com), Benjamin J. Henley (ben.henley@monash.edu)

**Abstract**

High-resolution palaeoclimate proxies are fundamental to our understanding of the diverse climatic history of the Australian

mainland, particularly given the deficiency in instrumental datasets spanning greater than a century. Annually resolved, tree-

ring based proxies play a unique role in addressing limitations in our knowledge of interannual to multi-decadal temperature

and hydroclimatic variability prior to the instrumental period. Here we present cross-dated ring-width (RW) and minimum

blue-intensity (BI) chronologies spanning 70 years (1929 – 1998) for *Podocarpus lawrencei* Hook.f., the Australian

mainland's only alpine conifer, based on nine full-disk cross-sections from Mount Loch in the Victorian Alps. Correlations

with climate variables from observation stations and gridded data reveal a significant positive relationship between RW and

mean monthly maximum temperatures in winter throughout central Victoria ($r = 0.62$, $p < 0.001$), and a significant negative

correlation to winter precipitation ($r = -0.51$, $p < 0.001$). We also found significant negative correlations between RW and

monthly snow depth at Spencer Creek in New South Wales ($r = -0.60$, $p < 0.001$). Of the assessed BI parameters, delta blue-

intensity ($\Delta$BI; the difference between early- and late-wood BI) displayed the greatest sensitivity to climate, with robust spatial

correlations with mean October to December maximum and minimum monthly temperatures ($r = -0.43$, $p < 0.001$; $r = -0.51$,

$p < 0.001$) and July precipitation ($r = 0.44$, $p < 0.001$), across large areas of northern Victoria. These promising findings

highlight the utility of this species for future work. With the very limited availability of suitable long-lived and cross-datable

species on the Australian mainland, these results have significant implications for advancing high-resolution palaeoclimate

science in southeastern Australia and for improving our understanding of past climate in the region.





**Plain text summary**

Tree-ring records provide a unique window into past climate variability. However, there are few such records from the Australian mainland. We present results from nine cross-sections of an alpine tree species from the Victorian Alps from 1929–1998. The tree ring widths have significant correlations with winter temperature, precipitation and snow depth. The intensity of reflected blue light from the wood surface shows a strong response to growing season temperature and winter precipitation.

**1 Introduction**

Documentation of climatic variations in the Northern Hemisphere (NH) is notably more comprehensive than that of the Southern Hemisphere (SH). Differences in the distribution of oceans and landmasses, as well as disparities related to cultural and historical development, continue to impair our understanding of SH climate (Villalba, 2000). A complete understanding of the climatic behaviour in a particular hemisphere is not possible without a thorough comprehension of the other (Pittock, 1978). That is, addressing the lack of SH climate knowledge is also a component of understanding NH and global climate.

The Australian continent encompasses a vast geographical extent with a diverse range of climate zones (Pittock, 2003). Instrumental and historical records of climate variables such as temperature and precipitation rarely extend beyond the start of the twentieth century (Bureau of Meteorology, 2001). Palaeoclimate proxies provide an important extension of the instrumental record, and aid in the development of meaningful assessments of the context of recent climate extremes and the fundamental nature of low-frequency climate variability. Dendroclimatology, the study of tree rings as a source of palaeoclimate proxies, has played an increasingly important role in our understanding of long-term climatic changes. Tree-ring studies have been widely applied, particularly in temperate environments, to produce centennial-scale climate information at annual resolution (*eg.* Villalba et al., 1996; Esper et al., 2002; Cook et al., 2006). However, progress in Australian dendroclimatology has been historically challenging due to the sparse availability of suitable materials and sites (Cook et al., 2006). Materials that exhibit annual growth rings are limited (Heinrich and Allen, 2013), and many do not live to sufficient ages for the development of multi-century records (Ogden 1978, 1981). Suitable environments for the preservation of subfossil material are also lacking in most parts of the Australian continent.

The Australian Alps constitute mainland Australia's alpine and subalpine regions. Tree species at their altitudinal threshold, such as those growing in high altitude and/or latitude sites, are typically highly sensitive to variability in climate, and therefore tend to best lend themselves to reconstructions (*e.g.* Villalba et al., 1994; Frank and Esper, 2005; Larocque and Smith, 2005). Dendroclimatological studies within the Australian Alps have focused on the widely distributed genus *Eucalyptus*, which exhibits clear annual rings at high elevations due to the strong growth limitation of winter temperatures (Brookhouse et al., 2008; Brookhouse and Bi, 2009). However, frequent mortality of specimens due to recurring fires in eucalypt habitats limits the availability of individuals of adequate age for long-term study. Given that continuous high-quality



climate records throughout the Australian Alps seldom extend beyond several decades, exploration of additional climate-
sensitive species with greater longevity and less affected by fire would be beneficial.

At high elevation, fire rarely penetrates into rock-scree environments . Due to the protection these environments offer
from fire, the age of vegetation growing within scree slopes can greatly exceed that of surrounding communities
(Schweingruber, 1992). In the alpine environments of New South Wales, Victoria, rock-scree sites often support pure stands
of *Podocarpus lawrencei* Hook f. (Williams et al., 2008). In these locations, *P. lawrencei* occurs as a procumbent shrub and
may attain an age of >500 years (Costin et al., 2000). Analysis of *P. lawrencei* revealed well defined, annual growth rings and
highly sensitive latewood bands, suggesting a promising opportunity for dendroclimatological study (Schweingruber, 1992).
However, attempts to generate chronologies for dendroclimatological analysis have been limited. *Podocarpus lawrencei*
exhibits highly eccentric (lobate) radial growth behaviour, with rings frequently affected by, or completely lost to, wedging.
These abnormalities make dating of core samples difficult, necessitating the collection of entire stem cross-sections. While the
destructive nature of collecting full cross-sections normally prevents their acquisition, sample materials became available
following widespread fires in 2002/03 in the Australian Alps. The severity of these fires meant previously protected stands of
fire-sensitive *P. lawrencei* were killed, allowing collection of full-disk cross-sections from multiple sites and an initial
investigation into their dendroclimatological potential (McDougall et al., 2012).

Although dendroclimatology has traditionally relied upon ring-width (RW) data, an array of alternative tree-ring
proxies also offer insights to climate histories. Maximum latewood density (MXD), for instance, represents the greatest density
incells formed at the latest stage of the growing season (Schweingruber et al., 1988). Maximum latewood density has been
widely used as a robust tree-ring proxy for growing-season temperature, particularly in the NH summer (*e.g.* Briffa et al., 1988;
D'Arrigo et al., 2000; Davi et al., 2003). However, the considerable cost and effort associated with generating MXD
chronologies has hindered their development and utilisation, especially in regions of the world in which dendroclimatology is
uncommon. The blue intensity (BI) technique – a recently developed approach that quantifies the intensity of blue light
reflected from a wood surface (McCarroll et al., 2002) – offers a cheaper and efficient surrogate for MXD (Björklund et al.,
2014; Wilson et al., 2014). Several experimental studies have demonstrated a strong, negative relationship between BI and
MXD, and sample preparation and generation of BI data can be performed at comparatively low expense (Campbell et al.,
2007, 2011; McCarroll et al., 2002; Björklund et al., 2014).

Although application of the BI method has been largely restricted to NH conifers, Brookhouse and Graham (2016)
conducted a preliminary assessment on the suitability of the BI method on *P. lawrencei* specimens from Mount Buller in
alpine NE Victoria (37.15ºS, 144.44ºE). They reported a highly significant correlation between the resulting BI chronology
and mean August-April temperature maxima ($r = -0.79$, $p < 0.0001$). The strength of this relationship greatly exceeded that of
RW. The BI method, then, may offer a superior source of climate-sensitive chronologies within the Australian Alps. Applying
this technique to Australian species may be the key to significantly improving our understanding of interannual to multi-
decadal climate variability prior to the instrumental period (Wilson et al., 2021). Together with existing palaeoclimatological





studies, an expansion of this work could provide a critical baseline for temperature and hydroclimate prior to the industrial era

and major land-use changes following European arrival.

This study will report a crossdated *P. lawrencei* RW chronology based on sampled material from a previously

unexplored site (Mt Loch) in the Victorian Alps (Fig. 1). Chronologies of multiple BI parameters will also be generated.. This

study will then investigate the sensitivity of the RW and BI chronologies to climate variability, and discuss the potential

contributions of *P. lawrencei* in the advancement towards building reliable, multi-centennial scale reconstructions for

southeastern Australia.

[Figure 1 here]

## 2 Methodology and data

### 2.1 Sampling site

The *P. lawrencei* samples employed in this study were collected from Mt. Loch (36.96ºS, 147.16ºE) in 2007. The sample site

comprises a steep, south-facing rock-scree slope at ~1800m elevation (Fig 2a). The climate at Mt. Loch, indicated by the

nearby (<2 km distance) Mt Hotham meteorological station, exhibits strong seasonality in temperature due to its high altitude

(Fig. 3a), and is characterised by cold winter conditions with consistent July to October snow cover (Wahren et al., 2001; Venn

and Morgan, 2007). Such environments host numerous *P. lawrencei* communities throughout the Australian Alps (McDougall

et al., 2012; Brookhouse and Graham, 2016). A total of nine stem cross-sections of up to 13 cm in diameter were examined in

this study.

[Figure 2 here]

[Figure 3 here]

### 2.2 Sample preparation

A transverse surface of each sample was initially flattened using a belt sander to produce a surface uniformly perpendicular to

the tree-ring boundaries. Prior to scanning, resins and other extractives were removed. Because the BI technique relies upon

reflected light, staining unrelated to wood formation can alter reflectance and associations with climate data. To overcome

these problems, resins and stains that discolour materials are extracted in a process that may exceed 30 hours for each sample.

These extraction processes typically rely on soxhlet apparatus and a hazardous extraction solution. Previous analysis of *P.*

*lawrencei* (see Brookhouse and Graham, 2016) refluxed radial laths in a soxhlet apparatus and ethanol/toluene solution for up

to 42 hours. As an alternative to soxhlet extraction, samples in this study were soaked in pure acetone. This method allows for

the preparation of entire disks, which is highly advantageous given the lobate growth behaviour of *P. lawrencei*. Preliminary

experiments using acetone treatment for resin removal (Frith, 2009) suggest a minimum required extraction time of 72 hours



for partially immersed 5-mm thick *Pinus sylvestris* L. cores. Subsequent applications of the same technique have revealed the

majority of extractives are removed from fully immersed samples after just 48 hours of treatment (Rydval et al., 2014).

Moreover, as an addition or alternative to standard BI methods, the difference between minimum and maximum BI (ΔBI) may

provide a data source that increases the climate sensitivity of BI data and eliminates the need for extraction.

A sub-sample of three discs was used in this study to assess the efficacy of acetone treatment on full cross sections

of *P. lawrencei*. Three samples, ranging from 7-13 cm in diameter and approximately 1 cm thick, were submerged in 100%

acetone in air-tight glass containers at room temperature for an initial 120 hour period, followed by an additional 48 hours of

immersion. Each sample was sanded to a 2000-grit (9.5 - 11.1 μm) finish after each extraction stage. When the samples surfaces

were free of scratches they were scanned on an Epson Perfection V850 Pro scanner using SilverFast Ai professional software,

at a resolution of 4800 dots per inch (dpi). An IT8 Calibration Target (IT8.7/2) was used to calibrate the scanner to ensure the

comparable reproduction of colours and brightness between scans (Campbell et al., 2011). After experimentation with different

soaking times, the remaining six cross-sections were soaked in acetone for the optimal 120 hour period prior to the development

of ΔBI, earlywood (EWBI) and latewood BI (LWBI) chronologies.

The highly lobate radial growth of *P. lawrencei* and extensive ring wedging made it necessary to measure multiple

axes of measurement from all samples. Ring-width (RW) measurements were produced using the program CooRecorder™

and visual crossdating was undertaken on CDendro™, with the additional aid of separate microscope magnification of the

wood surface and correlation analysis on the Dendrochronology Program Library in R (dplR: Bunn, 2008, 2009). Blue intensity

parameters (delta BI (ΔBI), earlywood BI (EWBI) and latewood BI (LWBI)) were measured along the same paths used for

our RW measurement. The Mt. Loch RW chronology was then crossdated against a remotely located *P. lawrencei* RW

chronology developed by Brookhouse and Graham (2016) from Mount Buller (67 km southwest of the Mt. Loch study site)

over a 79 year overlapping period (1906 - 1985, r = 0.72), to corroborate our dating.

**2.3 Chronology development**

Measured tree-ring series were detrended to remove sample-level noise prior to chronology estimation. Removing age-related

trends within RW series often involves the fitting of a negative exponential function (Hughes, 2011). However, growth

eccentricities associated with the lobate nature of growth in *P. lawrencei* means that a more flexible data-adaptive approach is

required. Smoothing splines equal to 67% of each RW series' length with a 50% frequency cutoff were applied to each

individual chronology in dplR (Bunn, 2008, 2010). Ring-width indices were calculated as residuals from the fitted curves. Due

to trends specific to each individual BI series, detrending the age-related growth trends of ΔBI, EWBI and LWBI chronologies

was undertaken in the same data-adaptive manner as the RW series. The robust bi-weight mean of the detrended residual series

were then calculated to produce the standardised RW and BI chronologies (Cook et al., 1990). The robust bi-weight approach

produces a chronology that is relatively unaffected by outliers - an important consideration in the study of *P. lawrencei* given

the highly eccentric growth behaviour and strong likelihood of outliers otherwise impacting the common signal (McDougall

et al., 2012). We further removed autocorrelation from the tree-ring indices within dplR. The pre-whitened (RES) chronologies





did not differ significantly from the non-prewhitened chronologies, and we therefore used the RES chronologies to assess
climate signals in the chronologies.

The quality and reliability of the chronologies were assessed using the expressed population signal (EPS), against the
generally accepted threshold of > 0.85 (Wigley et al., 1984). Additionally, due to a prevailing increasing trend apparent in the
BI RES chronologies, we produced first-differenced BI and RW RES chronologies for use in subsequent analysis with climate
data.

**2.4 Climate analysis**

The final chronologies were evaluated against  climate data spanning 70 years (1929 - 1998) due to constraints associated with
low sample resolution prior to the early 20th century. The relationship between the RW and BI chronologies and minimum and
maximum air temperature, precipitation, snow depth and streamflow data was then explored. Climate-correlation analysis was
conducted using observational data from the Bureau of Meteorology (BoM) station in Omeo, 41.8 km southeast of Mt Loch
(Fig. 3b). Continuous minimum and maximum monthly mean air temperature data from 1879 to 2009 are available at this site,
which correlates strongly with the substantially shorter dataset available in Hotham Heights (mean maximum monthly
temperature, r = 0.87; mean minimum monthly temperature, r = 0.71). It is important to note, however, the significant elevation
difference between our study location in Hotham Heights (~1800m) and the Omeo station (685m), and therefore the possible
variation in correlative strength of individual months. We further evaluated the sensitivity of our chronologies to mean monthly
precipitation data at Harrietville (Fig. 4)  - the closest station with sufficiently long records (data available from 1884 - 2015).
We further examined the relationship between our *P. lawrencei* chronologies and mean monthly snow depth records from
1954 - 2001 at Spencers Creek in NSW (~125 km northeast of Mt. Loch). Correlations with total monthly streamflow from
BoM hydrologic reference stations at Mitta Mitta River at Hinnomunjie and Livingstone creek at Omeo were also assessed.
See Table 1. for metadata pertaining to BoM observation stations used in this study. In addition to individual station data, we
explored relationships with the Australian Gridded Climate Data (Evans et al., 2020), which is a recent revision of the
Australian Water Availability Project (AWAP) gridded dataset (Jones et al., 2009). We investigated the spatial extent of the
relationship between RW and BI chronologies and mean monthly minimum and maximum temperatures and precipitation
across the alpine regions of Victoria and southern New South Wales, and further afield across Victoria. To ensure consistency
with the detrending approach for the RW and BI chronologies, we explored the links between interannual differences in both,
by applying first differencing to the climate data prior to correlation analysis.

[Figure 4 here]
[Table 1 here]



## 3 Results and discussion

This study presents nine successfully crossdated *P. lawrencei* specimens (13 individual series). Given the relatively small sample size, a strong common signal (exceeding the 0.85 EPS threshold) throughout a portion of the resulting RW chronology (Fig. 5) is encouraging. The RW chronology reached a mean EPS of 0.86 for the period 1929 - 1998. However, with individual specimen ages ranging from 67 to 327 years, future work with larger sample sizes would allow for the opportunity to utilise this species for climate analysis across multi-centennial time scales. Previous works have developed 114 year (McDougall et al., 2012) and 82 year (Brookhouse and Graham, 2016) RW chronologies from 48 and 13 *P. lawrencei* specimens respectively. Additionally, the ability to crossdate our RW chronology with a remotely located Mt. Buller chronology (Fig. 6a) (Brookhouse and Graham, 2016) demonstrates the spatial coherence in the sensitivity of this species to climate variables throughout southeast Australia. Such findings highlight the possible utility of *P. lawrencei* in the development of a strong dendroclimatic network throughout the Australian Alps.

[Figure 5 here]

[Figure 6 here]

## 3.1 Temperature correlations

The RW chronology shows increased variability from the 1900's, with particularly narrow rings observed in the 1950's and 60's. With regard to observation station data, the RW chronology response to mean monthly maximum temperatures from Omeo reveals positive, statistically significant ($p < 0.05$) correlations with the current growth season winter (June, July and August) and October and November in particular, and a strong, negative response to June to November temperature maxima of the previous growth season (Fig. 7a). Additionally, sensitivity of RW to mean minimum monthly temperatures at Omeo is dominated by statistically significant, negative correlations with previous growth season March to May minimum temperatures (Fig. 7b). The ability of the *P. lawrencei* RW chronology developed in this study to capture select months of growing season temperatures is consistent with previous dendroclimatological analysis of this species demonstrating air temperature as a dominant limiting growth factor (McDougall et al., 2012; Brookhouse and Graham, 2016). The influence of temperature throughout the growing season has previously been extensively documented as a primary control on the growth of coniferous species in high altitude and high latitude environments (eg. D'Arrigo et al., 1992; Brookhouse and Bi, 2009; Nishimura and Lovoque, 2011; Rydval et al., 2018). Concerning the strong, inverse relationship of the RW chronology to temperature of the previous growth season, similar response patterns have been found in other species such as lower elevation *Lagarostrobos franklinii* (Buckley et al., 1997) and the widespread *Phyllocladus aspleniifolius* (Allen et al., 2001) in Tasmanian, and high elevation New Zealand *Libocedrus bidwillii* (Palmer and Xiong, 2004). This may be related to a depletion of carbohydrate and nutrients reserves following a favourably warm growing season and accelerated growth rates.





[Figure 7 here]

Following the strong, positive correlation between RW and local temperature maxima during winter months, we

investigated the spatial signature of this relationship. The RW response to the AGCD was also dominated by a statistically

significant, positive correlation with mean June to August maximum temperatures (r = 0.62, p < 0.001), encompassing a broad

extent of central Victoria (Fig. 8a). This response is consistent with previously documented alpine *P. lawrencei* chronologies

(McDougall et al., 2012; Brookhouse and Graham, 2016). McDougall et al., (2012) reported a strong positive relationship

between RW and winter maximum temperatures, and a negative response to mean and maximum monthly snow depth. Given

the inverse relationship between winter temperature and snowfall, winter temperatures are suggested to reflect the magnitude

of winter snow depth and persistence of spring snow cover, which imposes significant impacts on vegetation growth (Kudo,

1991; Halter, 1998; Brookhouse et al., 2008). Snow cover is postulated to be a major determinant of the length of the phenology

and growing season of alpine vegetation (Kudo, 1991). With regard to Australian alpine flora, this sensitivity is consistent with

documented responses of *E. pauciflora* to winter snow cover (Brookhouse et al., 2008). The timing of growth initiation in

boreal and temperate environments is largely defined by temperature (Creber and Chaloner, 1984), with many coniferous and

deciduous species experiencing cessation in root growth at soil temperatures below 2 - 4°C (Halter, 1998). Persistent spring

snow cover due to cooler winter conditions delays the initiation of cambial activity (essential for the formation of wood cells)

and sustains such low soil temperatures, resulting in a shorter growing season and therefore a narrower growth-ring (Vaganov

et al., 1999; Kirdyanov et al., 2003). Additionally, considering the acute prostrate growth of *P. lawrencei* communities and the

likelihood of stands being buried by snow throughout winter, extended spring snow cover also presents a direct impediment

to wood production by delaying the commencement of photosynthesis (McDougall et al., 2012). Conversely, warm winter

temperatures are expected to accelerate snowmelt in spring, resulting in an earlier onset of photosynthesis and cambial

activation for *P. lawrencei*.

The ΔBI chronology developed in this study is most notably negatively correlated (with statistical significance) with

mean maximum temperatures in October, November and December of the current growth season (Fig. 7c). Taking the averaged

temperature maxima of these months produced a significantly strengthened correlation with the ΔBI chronology (Fig. 6b: r =

248   -0.43, p < 0.001). This response is comparable to the only previously constructed BI chronology for *P. lawrencei* by

Brookhouse and Graham (2016), whereby averaged August to April temperature maxima revealed the strongest BI-

temperature relationship. Moreover, ΔBI displayed a substantially stronger response to mean monthly minimum temperatures

throughout the same period (Fig. 6c: r = -0.51, p < 0.001).

[Figure 8 here]



The response of the ΔBI chronology to maximum and minimum temperatures throughout October to December is
consistent with many previous studies demonstrating that temperature of the growing period is the dominant climate parameter
influencing latewood density (eg. D'Arrigo et al., 2000; Davi et al., 2003; Kaczka et al., 2017; Blake et al., 2020). The
anatomical basis for wood density lies in the average amount and size of cell wall material within the tracheids (Vaganov et
al., 2006). During the growth season, tracheid size reduces, and density thereby increases, between earlywood and latewood
formation (Rathgeber et al., 2006; Cuny et al., 2014). An investigation into the interannual variability of wood density and
specific contributions of anatomical attributes in NH conifers by Björklund et al., (2017) found earlywood and latewood
density to be primarily influenced by tracheid size and cell wall dimensions respectively. Given that BI is an established proxy
measure of density, and assuming that it is recording similar variations in these structural anatomical properties (and that such
responses are consistent between hemispheres), the BI data in *P. lawrencei* may likewise reflect changes in tracheid size and
wall dimensions. Future studies of BI in *P. lawrencei* incorporating the exploration of interannual variations of these
anatomical properties could confirm this. Such work could further our understanding of the physiological controls on the BI-
density relationship, particularly as it relates to some SH species in which density variations have been noted to behave
differently to what is typically observed in NH conifers (Blake et al., 2020).

## 3.2 Precipitation, snow depth and streamflow correlations

Correlation analysis with precipitation data revealed a strong negative relationship between RW and June to November and
May precipitation of the current growth season, as well as a particularly strong positive response to precipitation in November
of the previous season (Fig. 9a). Additional correlations with AGCD most notably exhibited a significant negative relationship
between RW and mean June to August precipitation across southern Victoria and high altitude regions (Fig. 10a). Significant
negative correlations between the RW chronology and mean snow depth at Spencers Creek are also evident from June to
October (Fig. 11). This response is consistent with that observed in *P. lawrencei* RW from Mt. Blue Cow and Schlinks Pass
in NSW (McDougall et al., 2012), and reflects the spatial coherence of snow depth throughout the Australian alpine region.
Such results further demonstrate the previously discussed impact of temperature maxima on the persistence of spring snow
cover, and the consequent limitation on *P. lawrencei* radial growth. Additionally, given the significant contribution of snow
melt during winter and spring to the soil moisture balance in the Australian Alps (Costin et al., 1961), we postulate that the
positive response of RW to monthly snow depth of the previous growth season (Fig. 9) is related to excess moisture availability
providing optimum growth conditions the following growth year.

Relationships between BI and snow depth were non-significant (data not shown). However, the ΔBI chronology
exhibited particularly strong positive correlations with July mean monthly precipitation (Fig. 9b; Fig. 10c). Assessments of
the dendroclimatic potential of ΔBI for hydroclimatic reconstruction have thus far been limited. Notably, however, Seftigen et
al., (2020) recently reported an increase in the explained variance of a warm season, ΔBI-based precipitation reconstruction of
nearly 20 percentage points (to 55%), relative to the predictive skill of the RW-based reconstruction. The strong response of





RW and ΔBI in *P. lawrencei* to precipitation, as demonstrated in this study, hence emphasises the potential to improve the

coverage of high resolution, moisture sensitive proxy records in the Australian continent. This would present an opportunity

to produce new robust multi-century precipitation reconstructions for southeast Australia.

Tree-ring based reconstructions of additional hydrological parameters such as streamflow can provide valuable

insights to water resource managers and planners, particularly considering the confined range of observational records. In this

study, we therefore also conducted a preliminary evaluation of a potential *P. lawrencei* tree ring-streamflow relationship. The

lack of streamflow gauge datasets of comparable length to the *P. lawrencei* chronologies limited overlapping records to 43

295 years (1955 - 1998) for Mitta Mitta River, and just 26 years (1968 - 1994) for Livingstone Creek. Nonetheless, ΔBI exhibited

a particularly strong positive response to the current growth season June to March streamflow at Livingstone Creek and with

current season July streamflow at Mitta Mitta River (Fig. A1). Correlations between the RW chronology and streamflow data

at Livingstone Creek were non-significant. However, significant negative (positive) responses of RW to current (previous)

growth season streamflow in June and November at Mitta Mitta River are apparent (Fig. A1). Whilst some strong correlations

are present, the inconsistency of results across sites requires further explanation.

[Figure 9 here]

[Figure 10 here]

[Figure 11 here]

### 3.3 Limitations and future prospects

It is important to first note the necessity of full stem cross-sections for accurate dating of *P. lawrencei* due to highly eccentric

growth behaviour and frequent ring-wedging. Whilst the destructive sampling method required to obtain such cross-sections

may not always be justified or permissible (February and Stock, 1998; McDougall et al., 2012), ample opportunity exists for

further collection of fire-killed *P. lawrencei* stands throughout the Australian Alps, given the frequency of large-scale fire

activity in recent decades.

Both RW and BI chronologies in this study were inherently challenged by a limited sample size of just nine stem

cross-sections. Whilst the ΔBI data produced in this study presented a relatively strong common climate signal, multiple BI

chronologies reported in previous works have highlighted the requirement of a greater sample size to achieve comparable

interseries correlations to MXD (eg. Wilson et al., 2014; Blake et al., 2020). It is therefore likely that the common signal

strength of the *P. lawrencei* BI chronologies would increase substantially with the incorporation of additional series,

particularly on longer time scales.

Tree-ring parameters are known to comprise a considerable degree of non-climatic variance at lower frequencies

(Cook, 1985; Esper et al., 2005; Fonti et al., 2009; Björklund et al., 2020). Blue intensity chronologies in particular have

displayed stronger responses to temperature at high frequencies, yet generally poorer portrayals of low-frequency trends when



compared to RW (eg. Rydval et al., 2014; Wilson et al. 2021). Samples in this study consist of a vast range of ages. Whilst the smoothing spline method of detrending aims to preserve the majority of the resolvable low frequency variance (Cook et al., 1995), the extent to which this approach impacts the expression of low frequency variance requires further exploration with longer *P. lawrencei* chronologies.

Whilst correlation analysis between RW and BI chronologies and climate variables in this study has empirically highlighted the strength of the *P. lawrencei* growth response to climate, a more detailed understanding of physiological mechanisms is required to further establish the causality of these relationships. Given the limited study of the dendroclimatological properties of *P. lawrencei* thus far, and our relatively rudimentary understanding of the BI-density link, further sampling and physiological investigation is warranted. This would allow for better interpretation of RW and BI data, and improve upon an already encouraging expression of the climate signal.

Despite earlier quite pessimistic assessments of the Podocarpus genus for dendroclimatological purposes (Dunwiddie, 1979; February and Stock, 1998), due in part to limited sample availability (in the absence of fire-killed specimens) the strength of the observed correlations presented in this study of just nine samples are promising with regard to the future analysis of this species. The BI method appears to offer a promising additional proxy to RW for *P. lawrencei*, as in other SH species in which the climate signal of BI parameters has been explored (Blake et al., 2020; Wilson et al., 2021). The ongoing development and application of the BI method in *P. lawrencei*, particularly for longer, multi-centennial scale chronologies may help significantly improve our understanding of past climatic changes in the SH, given the valuable position annually resolved, tree ring-based proxies hold in palaeoclimatology.

## 4 Conclusion

Despite inherent challenges due to growth abnormalities, this study has presented crossdated *P. lawrencei* RW and BI chronologies on the order of 70 years for climate analysis (with individual (non-crossdated) series dating back to 1676), based on nine fire-killed specimens from Mt. Loch in the Victorian Alps. Ring-width measurements displayed the strongest responses to mean winter temperature maxima and snow depth, analogous to that demonstrated by high altitude *E. pauciflora* communities. The ΔBI parameter exhibited a greater sensitivity to climate than earlywood or latewood BI, presenting a particularly strong relationship with temperature and precipitation in the current growing season. This study offers encouraging results, particularly those pertaining to RW, for the increased utilisation of *P. lawrencei* in Australian dendroclimatology. With ongoing efforts to further reduce the limitations of the BI parameter and develop the most appropriate detrending methods, as well as the incorporation of anatomical analysis, the BI method also offers an important opportunity in Australian dendroclimatology. Given the known longevity of individual *P. lawrencei* specimens, the temporal extension and increased utilisation of *P. lawrencei* chronologies from the Australian Alps may help to provide an important perspective on climate change in the region. A detailed dendroclimatological network of this species could contribute meaningfully towards improving palaeoclimate data coverage in the Southern Hemisphere.



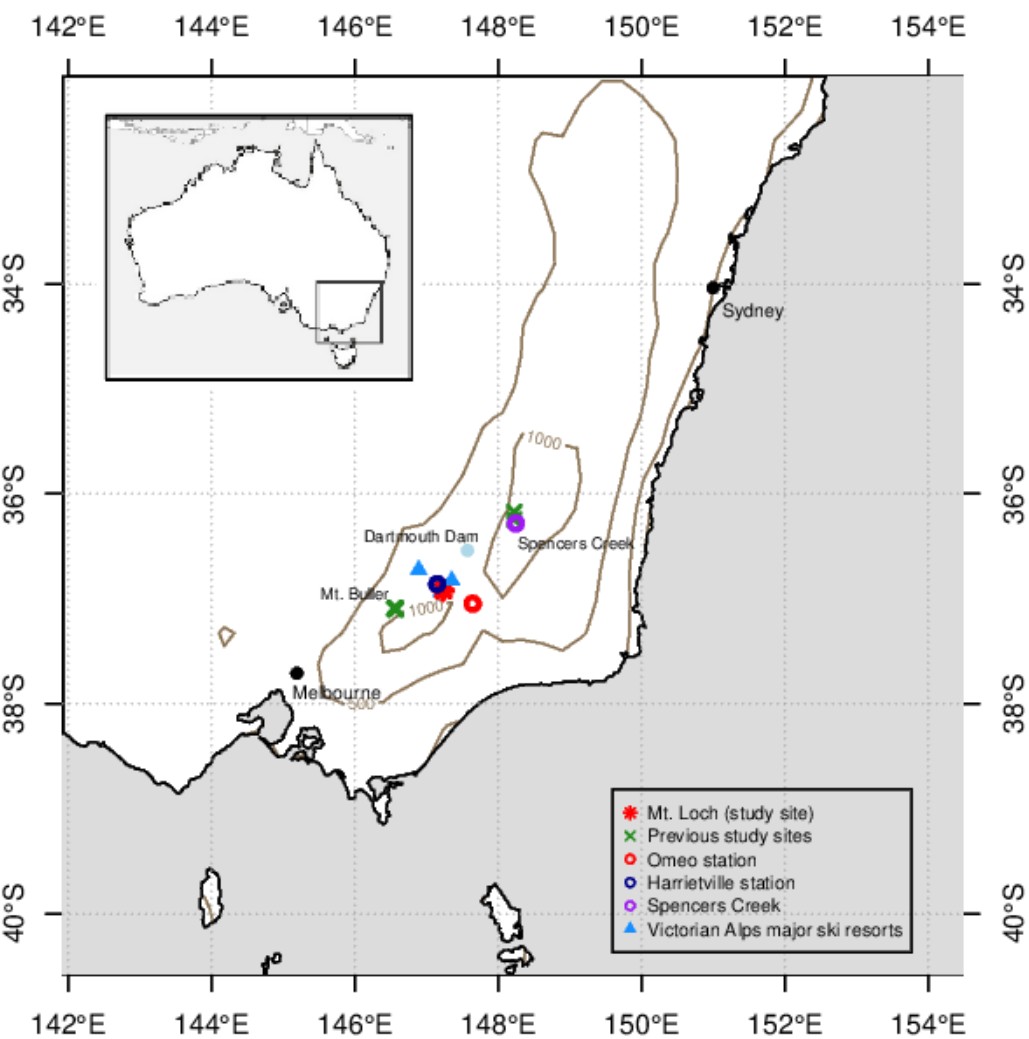

**Figure 1: Location of Mt. Loch sample site and main meteorological stations, and *P. lawrencei* study sites from previous works (Mt. Blue Cow and Schlinks Pass: McDougall et al., 2012, Mt. Buller: Brookhouse and Graham, 2016).**



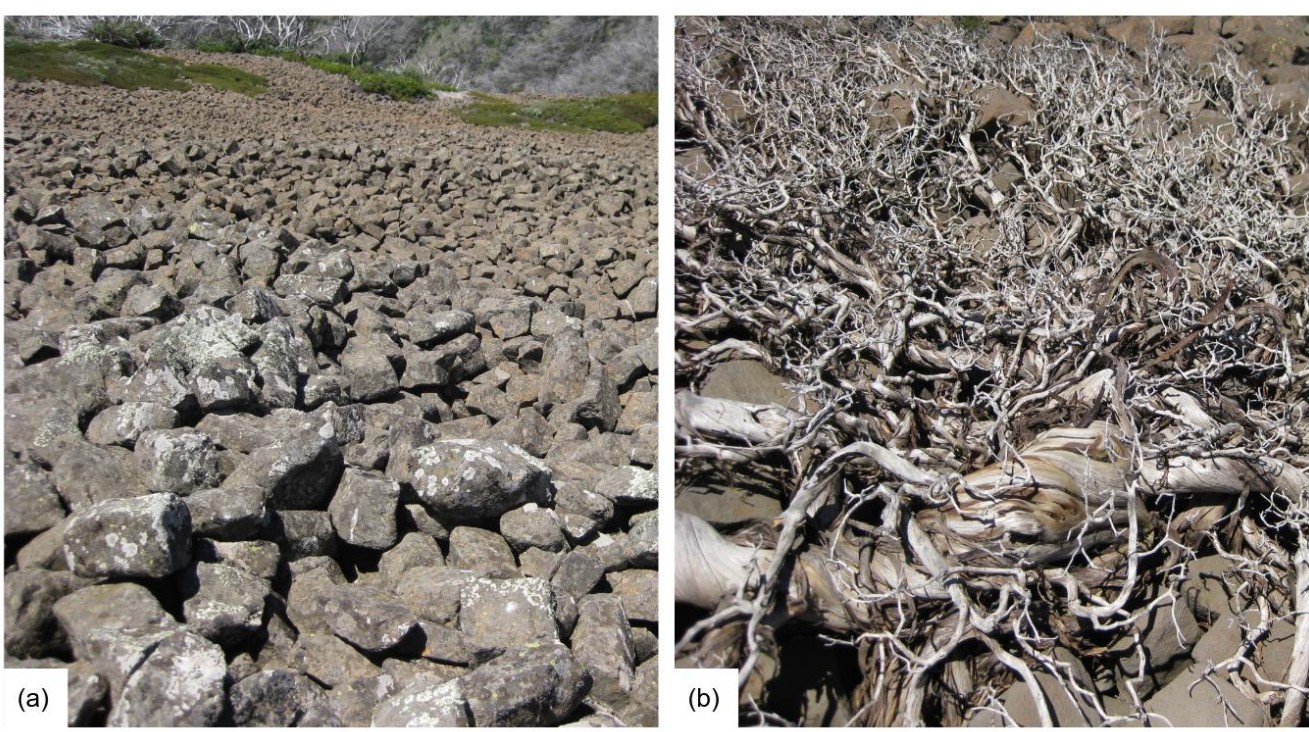

**Figure 2: (a) Mt. Loch boulder field on rock scree slope, from which samples for this study were collected. (b) Highly prostrate growth of fire-killed *P. lawrencei* stands from the same locality. Images taken by Matthew Brookhouse (ANU).**



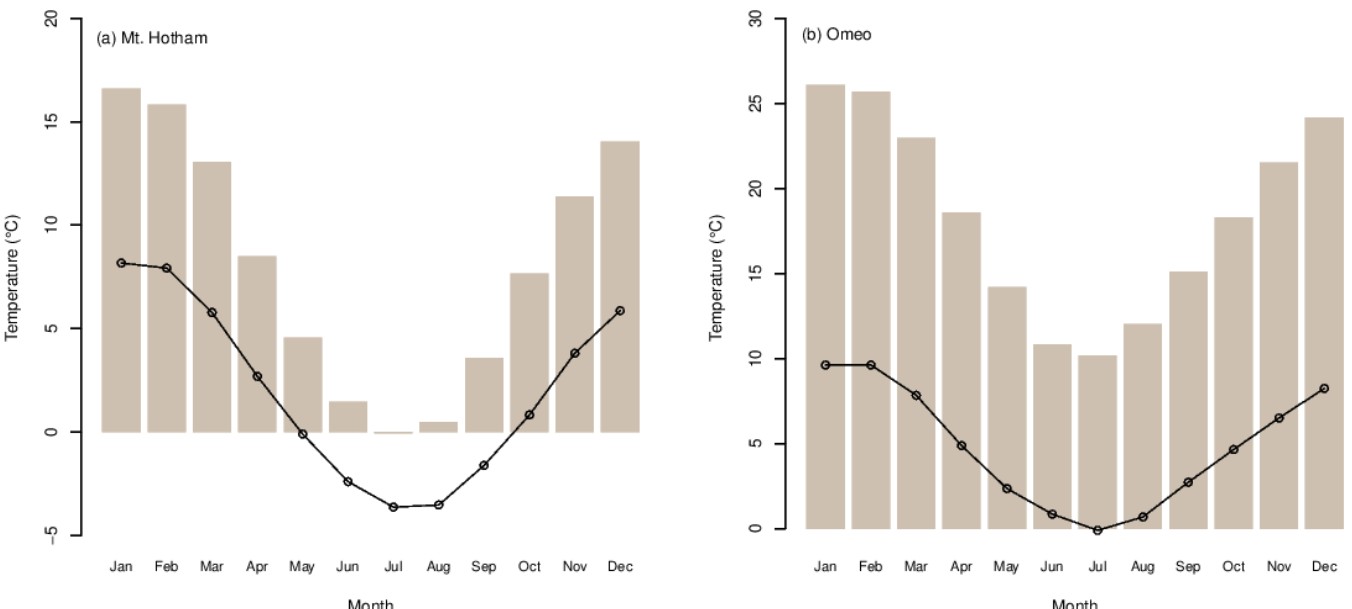

**Figure 3: Mean monthly maximum (shaded bars) and minimum (open circles) air temperature at (a) Mt. Hotham and (b) Omeo meteorological stations.**



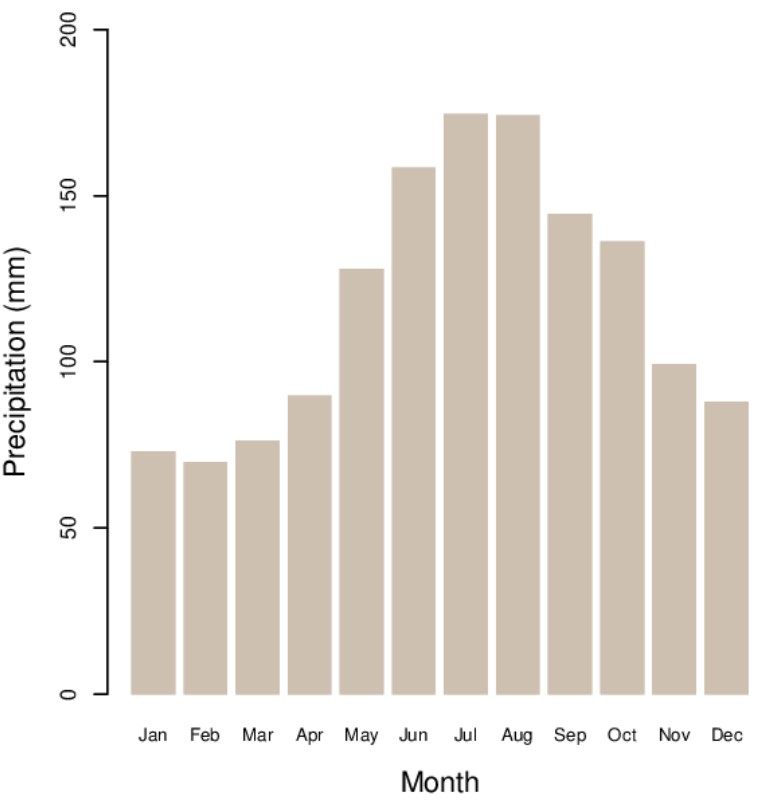

**Figure 4: Mean monthly precipitation at Harrietville meteorological station.**





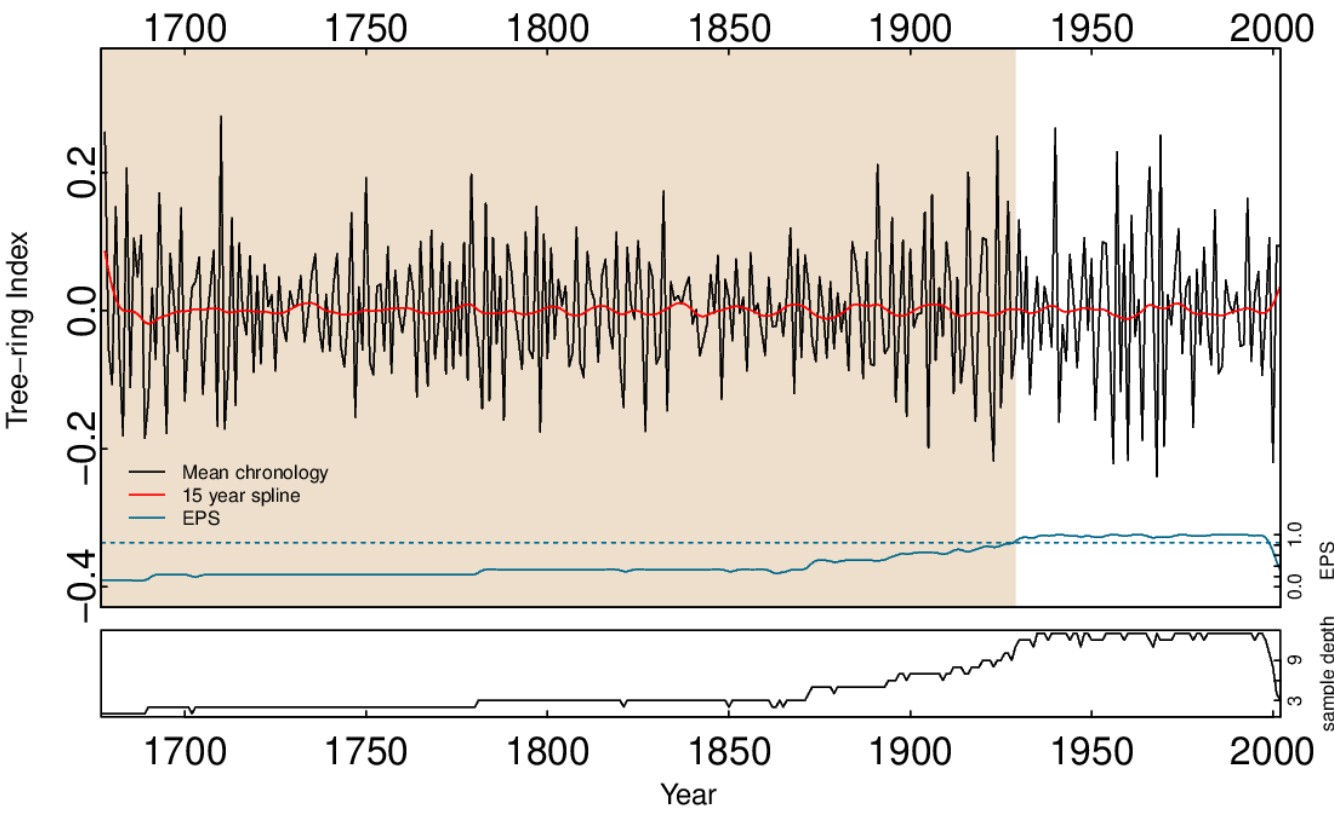

**Figure 5. Full mean ring width chronology (1676 - 2002) based on 13 Mt. Loch *P. lawrencei* series from 9 samples (13 individual series), with 30 year spline and two standard errors (top panel), and concurrent sample resolution (bottom panel). Expressed Population Signal (EPS) denoted by green line, with 0.85 threshold (dashed green horizontal line).**



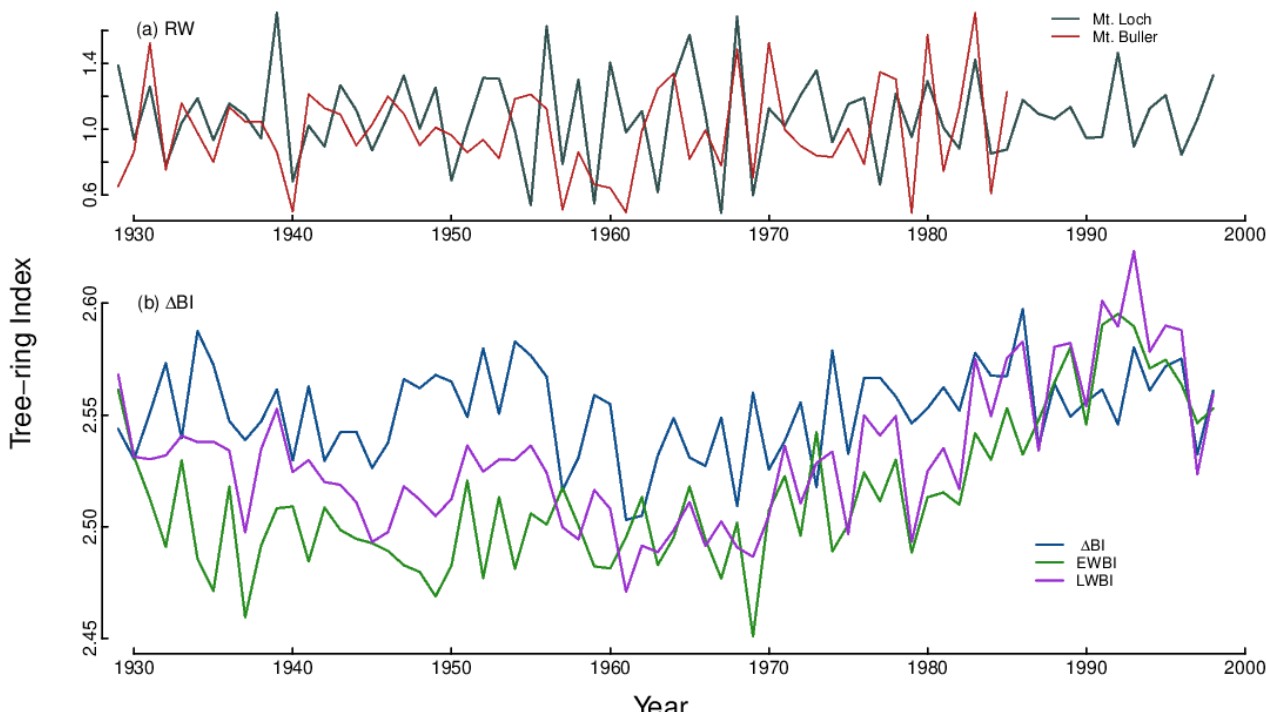

**Figure 6. (a) Detrended *P. lawrencei* ring width (RES) chronologies for Mt. Loch (this study) and Mt. Buller (Brookhouse and Graham 2016) and (b) earlywood and latewood BI chronologies, with the derived ΔBI parameter, for the 1929 - 1998 period.**



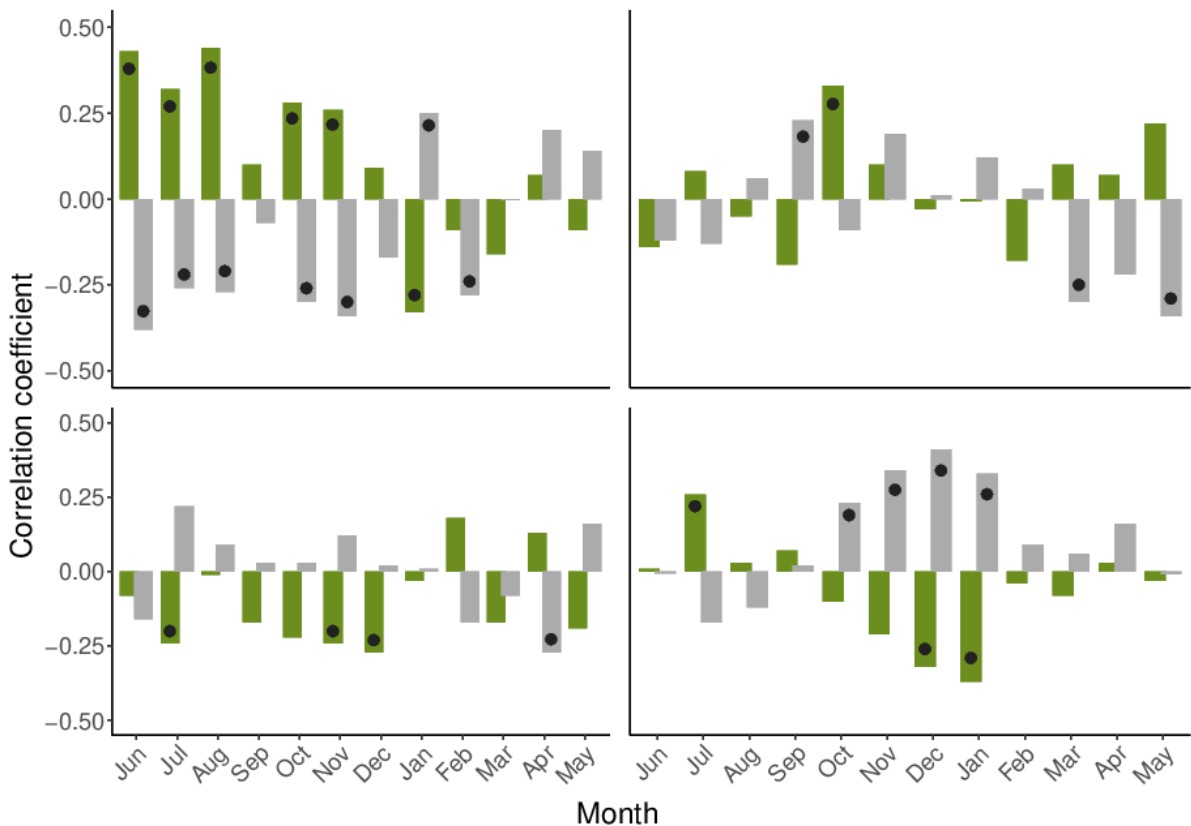

**Figure 7. Correlations between RW and ΔBI chronologies and mean minimum and maximum monthly temperature data from Omeo observation station across the period 1929-1998, for both the current and previous growth season. Black dots indicate statistical significance (p < 0.05).**



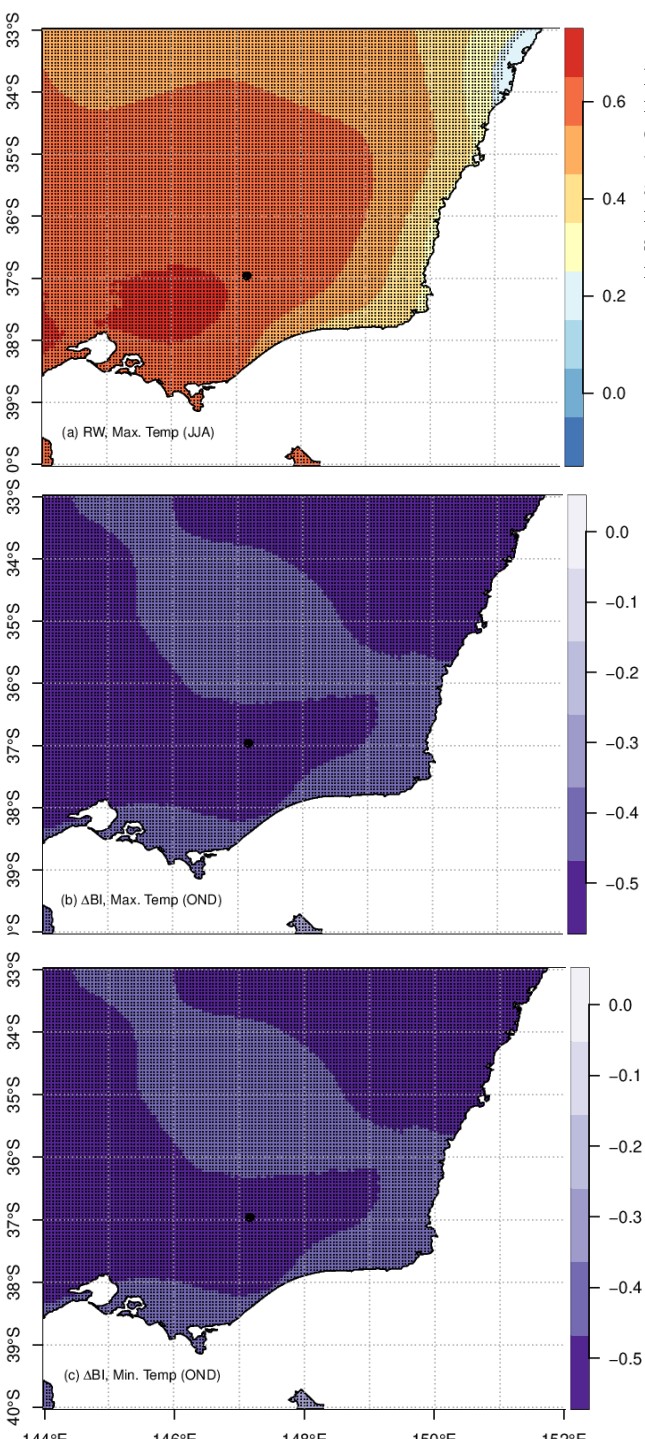

**Figure 8: RW and ΔBI chronology correlations with AGCD mean monthly temperatures data for 1929 - 1998 period. (a) RW correlation with mean June, July and August (winter) maximum temperatures, (b) ΔBI correlation with mean October, November and December maximum temperatures and (c) ΔBI correlation with mean December minimum temperature. Shaded areas represent statistically significant correlations (p < 0.05) and study site location is marked by black dot.**



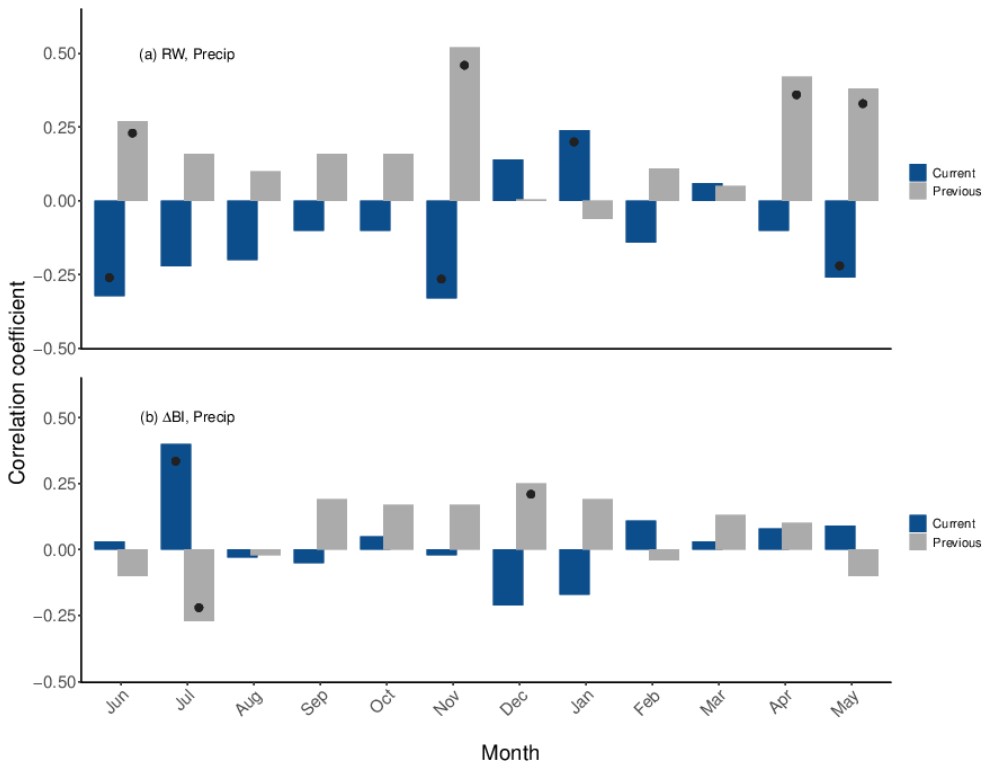

**Figure 9: Correlations between RW and ΔBI chronologies and mean monthly precipitation data from Harrietville observation station across the period 1929-1998, for both the current and previous growth season. Black dots indicate statistical significance ($p < 0.05$).**





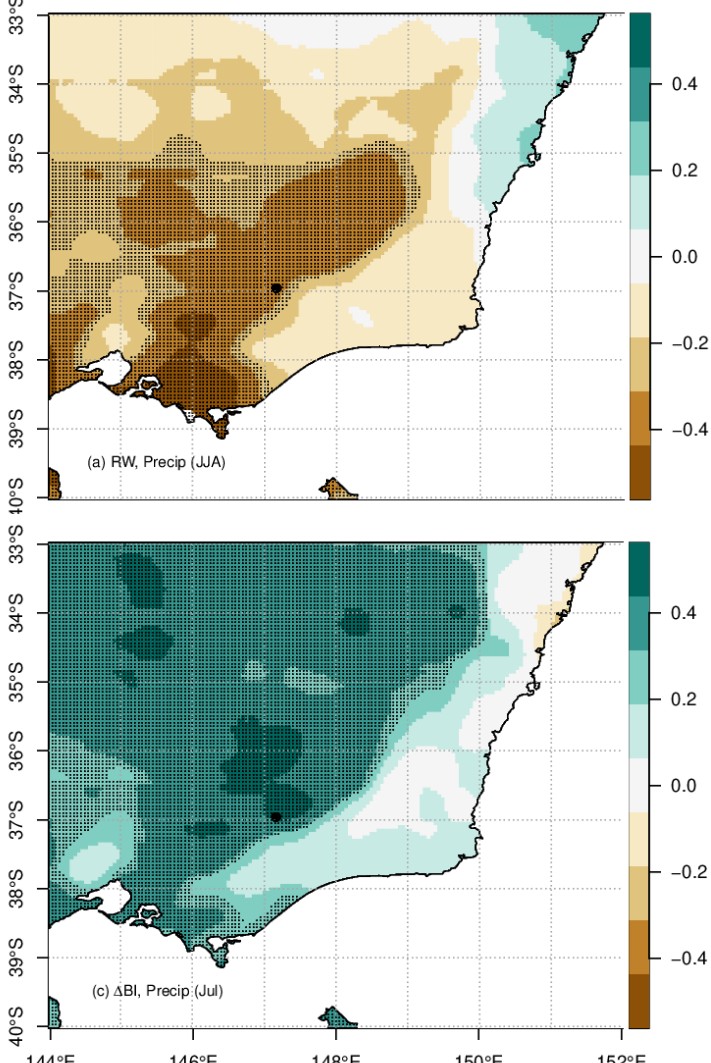

**Figure 10: RW and ΔBI chronology correlations with AGCD mean monthly precipitation data for 1929 - 1998 period. (a) RW correlation with mean June, July and August (winter) precipitation, (b) ΔBI correlation with mean July precipitation. Shaded areas represent statistically significant correlations (p < 0.05) and study site location is marked by black dot.**



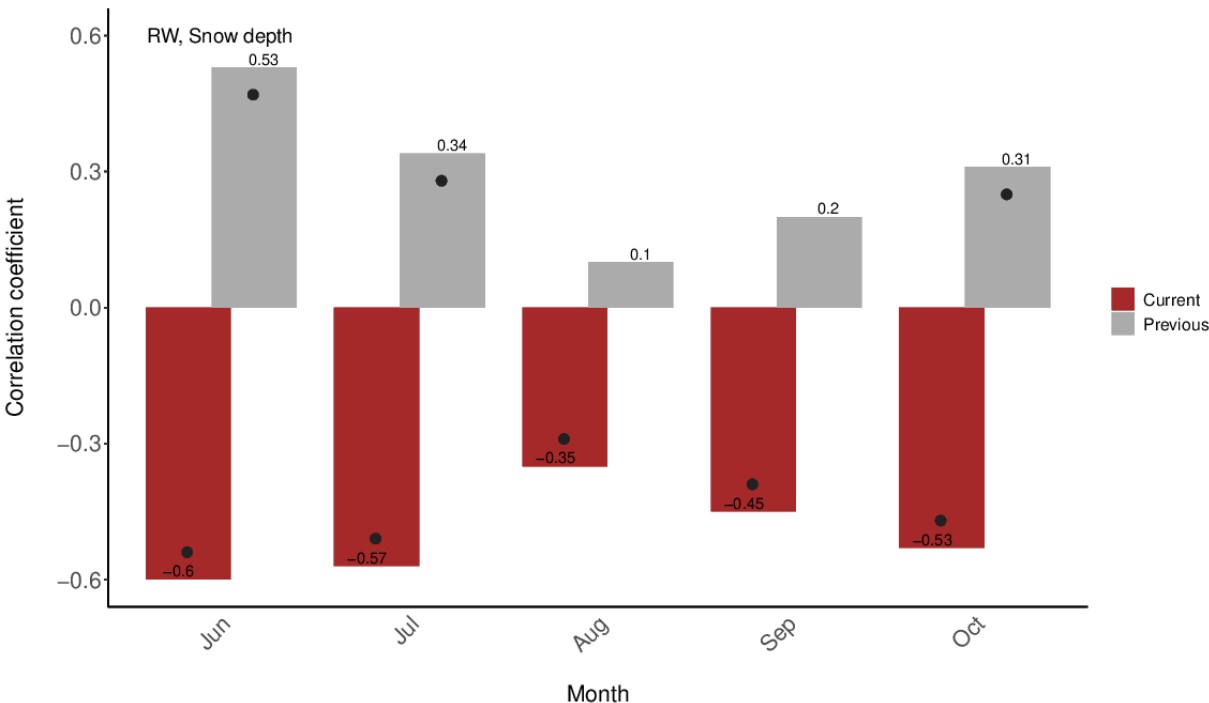

**Figure 11: Correlations between the Mt. Loch RW chronology and mean monthly snow depth at Spencers Creek, for the period 1954 - 1998. Black dots indicate statistical significance.**





| Station name | Station number | Latitude | Longitude | Elevation | Period of record | Variable(s) |
|---|---|---|---|---|---|---|
| Omeo Comparison VIC | 083025 | 37.10ºS | 147.60ºE | 685 m | 1879 - 2009 (130 years) | Mean monthly maximum and minimum temperature (ºC) |
| Mount Hotham VIC | 083085 | 36.98ºS | 147.13ºE | 1849m | 1990 - 2021 (31 years) | Mean monthly maximum and minimum temperature (ºC) |
| Harrietville VIC | 083012 | 36.89ºS | 147.06ºE | 396m | 1884 - 2015 (131 years) | Total monthly precipitation (mm) |
| Livingstone Creek at Omeo | 401209 | 37.11ºS | 147.57ºE | 691 m | 1968 - 1994 (26 years) | Total monthly streamflow (m$^3$/s) |
| Mitta Mitta River at Hinnomunjie | 401203 | 36.95ºS | 147.61ºE | 544 m | 1931 - 2021 (90 years) | Total monthly streamflow (ML) |

**Table 1. Bureau of Meteorology instrumental station metadata.**





**Appendix**



**Figure A1: Correlations between RW and ΔBI chronologies and total monthly streamflow (m³/s) at Livingstone Creek at Omeo for 1968 - 1994 period, and total monthly streamflow (ML) at Mitta Mitta River at Hinnomunjie across 1955 - 1998 period. Black dots indicate statistical significance ($p < 0.05$).**





## Acknowledgements

B.H. and J.O. wish to acknowledge the funding support of J.O. from B.H.'s Early Career Researcher Grant from the University of Melbourne. J.O. and B.H wish to acknowledge the support of the School of Earth, Atmosphere and Environment at Monash University during J.O.'s honours project. The authors would like to thank the Bureau of Meteorology for their provision of meteorological and hydrological data. We would also like to thank Johanna Spiers, James Pirozzi for their support for this project and Snowy Hydro Limited for their provision of meteorological and hydrological data. B.H. and J.O. further wish to sincerely thank Rob Evans for his advice, support and mentorship during the project. K.A acknowledges funding from ARC Future Fellowship FT200100102.

## Competing interests

The authors declare that they have no conflict of interest.

## Author contribution

J.O undertook all of the cross dating, measurements, and climate analysis, and took the lead role in writing the manuscript. B.H. conceived and developed the project, provided supervision, mentoring and equipment, and guided the climate analysis. M.B. provided the samples, conceived the study of the location and species and advised on the interpretation of the analysis. K.A. guided the dendrochronological analysis and interpretation. All authors contributed significantly to the manuscript.



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
