# Peer review of "Ring width and blue light chronologies of *Podocarpus lawrencei* from"

_Climate of the Past, 2022_

## Author Response (AR1)

Reference to changes made to the manuscript are in red text.

**Referee #1 Response**

1. **L 122-124. Please, explain the reason you don't follow this previous experience and takes less time in the extraction process.**

We explain the reasoning for this in the following sentence (L124-125): "This method allows for the preparation of entire disks, which is highly advantageous given the lobate growth behaviour of *P. lawrencei*."

2. **L 129-130. I understand that delta BI is used to correct the differences between heartwood and sapwood but not to correct stains. Please, specify what was the reason for calculating delta BI in this study. To correct color differences between heartwood and sapwood or to correct stains? Moreover, this sentence interrupts the continuity of the writing about the acetone treatment. Please, check it.**

The main reason was to attempt to correct the heartwood-sapwood colour difference. We shall clarify this in the transcript and integrate the sentence more clearly into the paragraph.

See L139-141

3. **L 133-134 Two treatments were applied? 1- 120 h acetone and 2- 168 h acetone? If what you wanted to evaluate is treatment efficacy, why not also evaluate the option acetone x 48 h and 72 h following the methodology applied by Rydval et al. (2014) and Frith, (2009)? It seems appropriate to mention briefly in this methods section, what was the optimal time you detected in this experimentation. Necessary information to continue with the treatment of the remaining 6 samples**

Thank you for the suggestion, we will be sure to highlight what the optimal treatment time was (120 hrs) in the methods section. And whilst we agree that it would have been optimal to follow the previous methodology of Rydval et al. (2014) and Frith (2009), unfortunately this was not an option due to strict COVID-19 lockdowns in Melbourne which left our samples stuck in the lab during the acetone soaking process for longer than expected.

See L148-150

4. **L 148. I assume that prior to developing the ring width and BI chronology, the series were crossdated. This is not explained in the manuscript so it is necessary to describe the crossdate process and report the statistics such as the mean series intercorrelation and mean sensitivity.**

We agree and shall include this information, including the key statistics mentioned.
See L152-156 and Table 1

5. **L 148. Please explain some where in method section the criteria used in the BI series. Which BI measurement is being considered? reflectance or absorbance?**

We will clarify that we are measuring BI reflectance.

See L155-156

6. **L 160-161. I suggest to plot the standard chronology also with its statistics (EPS, RBar, Mean sensitivity). Perhaps it can be included as supplementary material.**

We agree and shall include these, thank you for the suggestion.

See Table 1

7. **L 162. I suggest to report also the RBar statistic to complement the EPS.**

We agree and have included this in Table 1 which summarises the key chronology statistics.

See Table 1

8. **L 173-174. Please, clarify if the correlation values represent a particular monthly time series, a season or if it is the annual average of max and min monthly temperature. Also, indicate the common period between both weather stations**

The correlation values represent the annual average of max and min monthly temperature across the overlapping time period, we will clarify this. We will also include the overlapping period between the weather stations (1925 - 1975).

See L185-187

9. **L 176-177. Please explain how the chronology sensitivity to rainfall was evaluated. Which rainfall data were used? Did you consider months corresponding to previous years? how many?**

We evaluated the chronology sensitivity to total monthly rainfall at Harrietville, and looked at both the current growth year and the previous. We will make sure this is clearer in the methods section of the manuscript.

See L182-184 and L189-191

**10. L 191. One of the advantages that dendrochronology has over other climatic proxies is the high replicability. In this case the number of trees sampled is nine (13 individual series), which seems to me to be very poor for a representative chronology of the population growth. With such a low sample size it is highly probably that a lot of non-climatic noise will be introduced in the ring width chronology, especially if it is a species with lobular growth.**

We agree that this is a key limitation of our study, which was developed from an undergraduate thesis and was thus quite time-restrained. We acknowledge this in the discussion (L312-317) and the need for further sampling, however we believe that our results show a promising outlook for the future study of the species. Additionally our chronology surpassed the commonly used EPS threshold of 0.85 for the 70 year period (1929 - 1998) we used for climate analysis.

See L206-209 and L333-339

**11. L 208. Is winter part of the growth season of this species? normally, growth season occur since spring to autumn. Please, check.**

Climate correlation analysis was undertaken across a 12 month span, from June in the calendar year in which growth commenced, through to May of the following year in which growth ends. The wording of L208 ("current growth season winter") is referring to the winter months within the current 12 month (June-May) period, however we will amend this to just "current winter" as we agree that this sounds like we are including the winter month in the growth period.

See L224-225

**12. L 211. Significant positive correlations are also observed in October of the current period and in September of the previous period. Please, describe these results.**

We agree and shall be sure to mention this.

See L227-230

**13. L 212-214. Please, improve the wording of this sentence. example: The ability of our P. lawrencei RW chronology to capture temperature signals during some months of the growing season is consistent with previous dendroclimatological analysis of this species, demonstrating the air temperature as a dominant limiting growth factor.**

Thank you for the suggestion, we agree and will reword this.

See L230-232

**14. L 220-221. It is not clear enough to me the explanation why the species presents these antagonistic relationships between two consecutive years. During the current year it seems that it likes high winter temperatures to grow in spring-summer(?), while in the previous year the maximum winter temperatures negatively affect growth. Isn't the precipitation in June, November and autumn of the previous year also an important factor controlling tree growth?**

We explain in L220-221 that higher maximum temperatures in the previous winter may negatively impact growth due to a depletion of carbohydrates and nutrients reserves. This antagonistic relationship has been found in other Australian species, as listed in L217-220. Our results do show that precipitation in June, November and autumn of the previous year also have a significant influence on *P. lawrencei* growth, as noted in the following paragraph of the results/discussion section (L271-272).

See L235-239 and L292-293

**15. L 246. October is negatively correlated but not significant. Please, check.**

Thank you for pointing this out, we shall rectify this.

See L263-265

**16. L 246-248. I suggest to plot in fig. 6 the time series of max. temperature averaged for the months of Oct-Dec together with the delta BI series.**

We would rather use Fig. 6 to just show the chronologies, however we agree that this would be interesting to see and would be happy to include this as a supplementary figure.

See Figure A1

**17. L 250-251. What about the positive relationship between delta BI and minimum temperatures during the previous growing season? This relationship is even stronger than the one found during the current growing season.**

We agree, the current season min temperature relationship was highlighted as it was consistent with the results of Brookhouse and Graham (2016), but we will amend this section to include the results of the previous growth season.

See L268-271

**18. L 255. Please explain in methods what it means that BI is positive or negative. In results section, explain what it means that it correlates positively or negatively with temperature...what it would be indicating in physiological terms?**

We explain in this paragraph (L255-268), that since BI is used as a surrogate measurement for maximum latewood density (MXD), the relationship between BI and temperature reflects various changes in anatomical properties (however this relationship isn't fully understood). Regarding what it means for the BI-temperature correlations to be positive or negative: BI is negatively correlated with MXD and our BI data was not inverted. The strong negative relationships we observed between the BI chronology and temperature hence reflect the positive density-temperature relationships recorded in the NH. This response is also consistent with some SH species (eg: Celery-top pine and Huon pine (Wilson et al. 2021)).

See L275-277

19. **L 284. I suggest to use the total monthly precipitation value instead of the monthly mean value. The average is not totally representative of the monthly precipitation.**

We agree with this suggestion and will update the results using total monthly precipitation values instead.

Upon re-calculating the data it was found that we actually did use total monthly precipitation from both Harrietville observation station and the AGCD for climate analysis. This was erroneously labelled as mean monthly precipitation. Thank you for pointing this out, the figure captions have now been updated.

See Figure 8 and Figure 9

20. **Figures: Be consistent in the font size of the labels of all figures**

21. **Fig. 1 and 2. I suggest to combine figure 1 with figure 2, placing the map on the left and the two photos vertically on the right.**

Thank you for the suggestion, we agree and will do this.

See Figure 1

22. **Fig. 3 and 4. I suggest to combine Figure 3 with Figure 4. Placing the three figures in a vertical order since they share the same X-axis.**

We agree and will combine these figures also.

See Figure 2

23. **Fig. 6a. Please, indicate in the figure the correlation coefficient between the two ring width chronologies.**

We agree and shall add the correlation coefficient (r = 0.72) to the plot.

**24. Fig. 6b. The EWBI and LWBI show a strong positive trend. Any comments on this pattern? Does it have a climatic or biological explanation? Or is it a methodological artifact?**

It is plausible that the observed upward trend could still be due to colour differences (that may not be entirely visible) between the heartwood and the sapwood. However, it may also reflect the upward trajectory of temperature in recent decades - we will include the fig. 6 BI timeseries plot along with the time series of max temperature averaged for the months of Oct-Dec in supplementary as per the previous comment to assess this further.

**25. Fig. 7. Please, add the letters a,b,c,d corresponding to each of the 4 panels. Please indicate in the figure in a generalized way the vegetation growth period. I suggest to add in the panels the lines indicating the significance level, so that the reader can identify whether the correlation is highly or marginally significant.**

Thank you for pointing out this somehow overlooked error! We will add the letters to the figures and note the vegetation growth period in the figure caption for reference. We will also add lines to indicate the significance levels as suggested.

**26. Fig. 9. From a physiological point of view, it makes more sense to use the monthly total value instead of the monthly mean. The results can be very different if you use the monthly rainfall total instead of the monthly mean. I suggest that you perform the correlation analysis again using the monthly rainfall total.**

We agree with this suggestion and will re-calculate the results using total monthly rainfall instead.

As above: Upon re-calculating the data it was found that we actually did use total monthly precipitation from both Harrietville observation station and the AGCD for climate analysis. This was erroneously labelled as mean monthly precipitation. Thank you for pointing this out, the figure captions have now been updated.

**Referee #2**

1. **Title: I don't find the title particularly illuminating; it feels rather vague and not wholly accurate, in my opinion. I think more keywords from the abstract could be woven into the title. At the very least, I suggest making clear in the title that this is a dendro-based study on a new conifer species. Furthermore, this particular study does not report a "multicentennial record".**

Thank you for the suggestion - upon re-evaluating the title we agree that whilst the paper is "piloting" multi-centennial scale records, the use of this term could be misleading. We will change the title to the following to be more clear about the content of the paper: "Ring width and blue light chronologies of *Podocarpus lawrencei* from southeastern mainland Australia reveal a regional climate signal".

See L1

2. **Line 21: could you state in this sentence how far back in time the observations and gridded data extend?**

Both the observation station and gridded data extended back further in time than our final chronology length (1929 - 1998) and so we calculated the climate correlations across this entire 70 year period. We will clarify this in the sentence.

See L20-21

3. **Lines 51-53: I suggest expanding on this key point around limited progress in Australian dendroclimatology as this is critical justification for your research and will better accommodate the broad audience of the journal.**

Thank you for the suggestion, we agree and shall develop this point further in the introduction.

See L44-57

4. **Lines 57-58 and 65-71: The paucity of long-term reconstructions and the benefits of high elevation areas are emphasised but some commentary on the representativeness of alpine climate reconstructions for the wider region would be useful.**

We agree that this would be useful additional information and will expand upon this also.

See L58-69

5. **Line 112-119: I have modest experience of dendroclimatology but, with that caveat in mind, nine specimens feels rather limited. I suggest some justification (how**

**does this compare to similar case studies like McDougall et al. 2012 and Brookhouse & Graham 2016?) and critical reflection on the appropriateness of this sampling strategy and any limitations this may introduce would be useful at this stage.**

We agree that more specimens would certainly be ideal. As this study was undertaken as an undergraduate thesis, time was quite restricted, and we acknowledge the low sample size as a key limitation in L312-317. Whilst our chronology is comprised of less individual specimens than McDougall et al. (2012) (48 samples) and Brookhouse and Graham (2016) (13 samples), we believe that our study still holds the novelty of consisting of a new *P. lawrencei* site in which we explored the spatial signature of climate sensitivity, as well as the application of a simpler and cheaper resin extraction technique. Additionally, our chronology exceeded the commonly used EPS threshold of 0.85 for the 1929-1998 period, and this was hence the portion of the chronologies that we used for climate analysis. We agree with the need to further justify the viability of the sample size and will emphasise this information in the discussion.

See L106-111 and L206-216 and L333-339

6. **Second, could the authors clarify the method(s) used to physically obtain the crosssections? Were these sliced out of the trunk or did they study stumps? Or did they repeat the approach outlined on Lines 76-77 from McDougall et al. 2012? I'm guessing the latter but this should be made clear in the Methods section.**

The latter is correct and we will add this information to the methods.

See L116-118

7. **Lines 140, 151, and elsewhere: The authors repeatedly highlight the highly lobate radial growth of P. lawrencei. I – and I suspect other readers – would find a photo of one of the cross-sections very useful.**

We agree that an image would be helpful, and will include a scan of one of the cross-sections to illustrate the lobate growth behaviour.

See Figure 3

8. **Lines 182 – 185: I suggest providing a bit more technical detail on the gridded climate data, especially the temporal and spatial scale of those datasets.**

We agree with this suggestion. The AGCD extends from 1900-2020 with a grid averaged resolution of 0.05 degrees (approximately 5km): we will include this information.

See L195-198

9. **Lines 195-196: I'm a little surprised that these two dendrochronological studies (McDougall et al. 2012 and Brookhouse & Graham 2016) are not cited in the introduction. Given the important claims made in this paper about novelty, being more transparent about what research has already been conducted and how your study builds upon existing work is important. The point about developing a strong network (Lines 199-200) is valid but this secondary aim could be stated earlier in the paper.**

Whilst we mentioned Brookhouse and Graham (2016) in L89-93 in the introduction when discussing the previous application of the BI technique on *P. lawrencei,* we agree that the importance of these two studies is understated in the introduction as a whole, and we will further emphasise how our study builds on these earlier works. We also agree that the aim of developing a future dendroclimatological network ought to be mentioned earlier, and will include this in the introduction also.

See L106-111

10. **Results and Discussion: Whilst I appreciate the focus of this paper is testing for climate signals, I'm surprised more detailed reporting of the cross-dating and chronological development is not presented. The full chronology is shown on Figure 5 but minimal corresponding text is presented in Section 3. The authors state on Line 94 that specimen ages range from 67 to 327 years – this could be elaborated upon, especially to provide deeper justification for your decision to start the analysis from 1929 due to "low sample resolution" Line 169.**

Thank you for the suggestion, we agree that in an effort to be concise we have omitted some helpful information detailing the chronology development, and will expand on this in the methods. Regarding our justification for conducting correlation analysis of climate variables on the 1929-1998 portion of the full chronology, the running mean of the EPS statistic in figure 5 gives an indication of our chronology reliability. The sample depth panel in this figure shows that at least 11 radii were required to meet the EPS threshold of 0.85, and this occurs from 1929-1998.

See L151-178 and L206-216

11. **Similarly, given the analysis is on 20th-century reconstructions, a refined Figure 5 focusing on only that time window would be useful, perhaps as a second panel. In its current form, statements like Lines 206-207 "particularly narrow rings observed in the 1905s and 60s" are tricky to pick out by eye.**

Whilst figure 5 shows the full RW chronology, figure 6 shows the RW and BI chronologies, focusing on just the 1929-1998 period. We will include a figure reference to this plot in the sentence at L206-207 so that readers are directed to the close up of that time window.

See L223-224 and Figure 5

**12. Lines 228-244: I found this analysis to be fascinating! I know little about this species but Te justification for a strong relationship between winter temperature and RW presented in this segment is convincing.**

The dominant effect of winter temperature and snow cover on *P. lawrencei* RW is certainly a key result! Exploring the spatial extent of this relationship with the AGCD was interesting also - very encouraging, particularly given the sample size.

**13. Figure 1 appears rather blurry on my screen. This can be easily rectified but I felt worth flagging with the authors.**

Thank you for making note of this, we will check this.

See Figure 1

**14. Figure 5: In addition to the comments above about the data plotted in this figure and the figure itself, it's not clear how the standard errors have been visualised on the top panel? I'm also unclear what the y-axis label on the bottom panel refers to – what is "sample depth"?**

Thank you for pointing this out. The standard errors were previously included in the graph but were removed, and the caption was not updated - this will be rectified. The bottom panel refers to the sample resolution: as the series are of different lengths we included this information to illustrate the number of tree-ring series used throughout the chronology above.

See Figure 4

**15. Figure 7: could you clarify in the caption which parameter is represented by the green and grey bars?**

This error will be rectified also to include information about what each panel represents, apologies!

See Figure 6